# Examining the Prevalence, Nutritional Quality and Marketing of Foods with Voluntary Nutrient Additions in the Canadian Food Supply

**DOI:** 10.3390/nu13093115

**Published:** 2021-09-05

**Authors:** Anthea Christoforou, Sheida Norsen, Jodi Bernstein, Mary L’Abbe

**Affiliations:** Department of Nutritional Sciences, Temerty Faculty of Medicine, University of Toronto, Medical Sciences Building, Room 5368, 1 King’s College Circle, Toronto, ON M5S 1A8, Canada; Sheida.Noorhosseini@mail.utoronto.ca (S.N.); jodi.bernstein@mail.utoronto.ca (J.B.); mary.labbe@utoronto.ca (M.L.)

**Keywords:** food fortification, fortification policy, discretionary fortification, voluntary fortification, supplemented foods, functional foods, Canada, food marketing

## Abstract

Foods with voluntary nutritional additions are a fast-growing sector of the global food industry. In Canada, while the addition of nutrients to foods has been regulated through fortification regulations, parallel policies which aim to encourage product innovation have also allowed for the voluntary addition of nutrients and other novel ingredients to ‘supplemented’ and ‘functional’ foods. Concerns have been raised that the consumption of these products may have negative repercussions on population health, such as high nutrient intakes inappropriate for certain population subgroups (e.g., children) and the shifting of dietary patterns to include more unhealthy foods. The aim of this study was to evaluate the prevalence, nutritional quality, and marketing characteristics of foods with added nutrients in the Canadian market. We found many nutritionally-enhanced foods contained high levels of nutrients beyond recommended intakes, despite these nutrients having no evidence of inadequacy in the Canadian population. Additionally, a large proportion of foods with added nutrients had poor nutrient profiles (were deemed ‘less healthy’ than their non-enhanced counterparts) and carried heavy marketing on their labels, regardless of their nutritional quality. Taken together these findings raise concerns about foods with voluntary nutrient additions and suggest the need to further investigate consumer attitudes and decision-making towards these foods.

## 1. Introduction

In Canada, the fortification of foods with nutrients, such as vitamins and minerals, has traditionally been tightly regulated, used as a means to prevent or correct nutrient deficiencies and their related morbidities in the population [1]. The Canadian *Food and Drug Regulations* (*FDR*) defines two types of fortification. Mandatory fortification—which requires that certain micronutrients be added to specific foods, such as the addition of vitamin D to milk—and voluntary fortification—which allows for a range of micronutrients to be voluntarily added to certain foods, such as the addition of B vitamins to breakfast cereals [2]. While historical micronutrient deficiencies have been mitigated through mandatory fortification programs, foods with voluntary nutrient additions (i.e., vitamins, minerals, amino acids, bio-actives and other novel ingredients), have gained market access and are a fast-growing sector of the Canadian food industry. Both mandatory and voluntary fortification, permit vitamins and minerals additions to foods only in amounts based on the highest Recommended Daily Allowances (RDAs)/Adequate Intakes (AIs) and Tolerable Upper Levels (ULs) within the population, typically values indicated for males 19 years of age and older [3,4] There is concern that this method of setting maximum levels may lead to excessive micronutrient intakes in certain segments of the population, such as children [5,6,7], because voluntarily fortified foods, such as breakfast cereals and fruit juices, tend to be marketed to all members of the population [5].

The voluntary addition of nutrients and other novel ingredients to foods has also become possible through parallel policies that aim to, in part, encourage product innovation [8,9]. For instance, Supplemented Foods (SFs) are nutritionally-enhanced products containing high levels of added vitamins, minerals, amino acids, or caffeine that do not comply with traditional fortification and enrichment policies. To bypass more restrictive policies imposed under the food regulatory framework, many SFs were previously classified as Natural Health Products (NHPs) and regulated as drugs. However, as the introduction of food-like NHPs began to flourish, Health Canada recognized that many would be more appropriately classified as foods under the FDR since they were being packaged, marketed, and consumed as conventional food products [10].

Additionally, Functional Foods (FFs), or foods containing added ingredients that are associated with providing a physiological benefit and/or reducing the risk of chronic disease—such as novel fibres, protein isolates and concentrates, pro- and prebiotics, and omega-3 and omega-6 fatty acids—have been gaining popularity in the market [11]. However, there is currently no regulatory definition or list of ingredients that a food must contain to be deemed ‘functional’.

A series of guidance documents have been published by Health Canada [8,9,12,13] to provide food manufacturers with information on how SFs and FFs may be formulated, including minimum and maximum levels for the addition of certain nutrients, such as caffeine. However, to date, there is no separate regulatory framework or product identifier to distinguish SFs and FFs from other conventional foods. As a result, consumer confusion may arise surrounding the appropriate use of these foods within the context of a healthy, balanced diet. Researchers and health professionals are also concerned that the consumption of foods with voluntary nutrient enhancements may have detrimental effects on population health, resulting in high intakes of certain nutrients that exceed upper levels (ULs) and/or the overconsumption of certain substances [14].

There are also concerns that manufacturers may use voluntary nutrient additions or enhancements as a marketing strategy, adding indiscriminate amounts of vitamins, minerals and other novel ingredients to foods that are less healthy to increase sales [7,14]. Although the FDR enforce specific regulations surrounding the use of certain claims, such as nutrient content and health claims [15], Canada currently does not require foods to have an overall healthy profile to carry a claim and there are no specific regulations which govern the use of other on-package nutritional promotion such as front-of-package systems or symbols (FOPS).

While some studies have been conducted on the health outcomes of voluntary fortification in other countries [6,16], there is currently a paucity of data on the extent and nature of food and beverage products with voluntary nutrient additions in the Canadian marketplace. The objective of the current study was therefore to assess the prevalence and nutritional quality of foods with voluntary nutrients additions, including SFs, FFs and foods with very high levels of voluntary fortification (VHVMs) in Canada, and to secondarily examine the degree of nutritional marketing found on these products.

## 2. Materials and Methods

### 2.1. Data Collection

#### Food Label Information Program 2013 (FLIP 2013)

Data was acquired from the University of Toronto Food Label Information Program 2013 (FLIP 2013), an online database containing 15,401 pre-packaged foods and beverages from the top four supermarket chains in Canada—Loblaws, Sobeys, Metro, and Safeway—and representing 75.4% of the grocery retail market share [17]. FLIP 2013 provides a cross-sectional overview of the nutritional composition and on-package marketing of pre-packaged products in the Canadian marketplace in 2013. Complete details of the FLIP 2013 methodology can be found in Bernstein et al. (2016) [18]. Products with errors in manufacturer labelling (*n* = 10), meal replacements, foods intended for children under the age of 4, and products lacking a standard Canadian Nutrition Facts table (NFt) were excluded from this analysis, leaving a final sample of 15,332 products. The NFt and Ingredients List of all remaining foods and beverages were examined to obtain information on the type and amount of added ingredient in each product. All foods were categorized based on food subcategories in Schedule M of the FDR [19]. Schedule M is a component of the Canadian Food and Drug Nutrition Labelling Regulations [B.01.001] and lists serving size reference amounts and recommended serving size ranges for food categories and subcategories.

### 2.2. Data Analysis–Classifying Foods and Beverages with Voluntary Nutrient Additions

#### 2.2.1. Food and Beverages with High Levels of Voluntary Fortification

For the purposes of this study, voluntary fortified foods were those with very high levels of voluntary fortification (VHVM), containing a vitamin or mineral in an amount that is greater than 25% of the Daily Value (DV) in accordance with the regulations pertaining to voluntary fortification in the FDR [10]. The percent DV is displayed on the NFt and is a guide to the nutrients in one serving of food. It is based on a 2000-calorie diet for healthy adults [3]. Foods and beverages that did not contain added nutrients, contained added nutrients in lower amounts (≤25% of the DV), or contained added nutrients for purposes other than voluntary fortification (e.g., mandatory fortification or enrichment, or food additive purposes) were excluded from this group.

#### 2.2.2. Supplemented Foods

At the time that these analyses were conducted, Health Canada had not yet finalized its definition of a SF. Therefore, for the purpose of this study, SF were defined as foods that contain added vitamins, minerals, amino acids, or caffeine, as indicated in the Ingredients List, added in amounts other than that which is permissible by the current FDR for fortification or enrichment purposes. SF with vitamin/mineral additions that exceeded the RDA or AI were also identified.

The amount of vitamin, mineral, or amino acid present in a SF was calculated using percent DVs provided in the NF and analyzed per stated serving size. The resultant levels were then compared to the highest Recommended Dietary Allowances (RDAs)—or Adequate Intakes (AIs) if RDAs were not available—and Upper Limits (ULs) of children and adolescents (4–13 years) and adults (19 years and older) within the population, excluding pregnant and lactating women, and individuals over the age of 70 [4].

#### 2.2.3. Functional Foods

In accordance by the definition provided by Health Canada, FF were considered to be those which contained nutritional enhancements and made some indication, in the form of claims or other statements, that they contained these added substances for the purpose of providing a health benefit [11]. Foods that contained a functional ingredient but did not make any claim or statement regarding its intended physiological benefit were excluded, since in such cases, manufacturers may have added these substances to foods for other purposes (e.g., acacia gum added as a food additive). The type of functional ingredient added was characterized as: herbals/bioactives, novel fibres, omega-3/omega-6 fatty acids, protein concentrates/isolates, and other novel ingredients. The amount of functional ingredient added to a food, however, could not be determined from the information provided in the NFt or Ingredients List and therefore was not calculated.

### 2.3. Calculating Nutritional Profiling Scores

To examine a products overall nutritional quality, the validated Nutrient Profiling Scoring Criterion (NPSC) system, created by Food Standards Australia New Zealand (FSANZ) [20], was used to create a nutrient profile score for each product. The FSANZ model separates foods into three categories: beverages (Category 1); cheese, edible oil, edible oil spreads, butter and margarine (Category 3); and any food other than those included in categories 1 and 3 (Category 2). A score is then created for each food or beverage according to a points-based system relying on category-specific nutrient content thresholds. Points are added for ‘negative’ nutrients, such as energy content, saturated fat, total sugars, and sodium, and points are deducted for ‘positive’ nutrients, such as fruit/vegetable/nut/legume, protein, and fibre content. Scores fall on a range of −18 to 81, with lower scores indicating ‘healthy’ foods and higher scores indicating ‘less healthy’ foods. Foods in categories 1, 2, and 3 are eligible to carry health claims under FSANZ only if their final scores are less than 1, 4, or 28, respectively.

### 2.4. Identifying on-Package Marketing

All nutrition-related marketing on FLIP products were recorded and classified into several categories, including regulated claims (i.e., nutrient content claims, disease risk reduction claims, and function claims) and unregulated FOPS. Each regulated claim or FOPS was counted individually as one marketing item. The number of regulated claims, FOPS, and total marketing items occurring on foods with voluntary nutrient enhancements were compared to similar products without such additions. If a product repeated the same claim or FOPS multiple times on different panels, the claim or FOPS was counted only once.

### 2.5. Statistical Analyses

All statistical analyses were performed using SAS version 9.3 (Statistical Analysis Software Co, Cary, NC, USA). Categorical variables were reported as percentages and frequencies and continuous variables are quantified as medians, Q1s, and Q3s. The Wilcoxon signed rank test was used to determine statistical differences in median NPSC scores or marketing items amongst SFs, FFs and VHVMs and non-nutritionally enhanced counterparts. Chi squared tests were used to compare the proportions of SFs, FFs and VHVMs to non-nutritionally enhanced counterparts meeting ‘healthy’ cut-points based on the FSANZ NPSC system to be eligible to carry health claims. Only food subcategories containing at least 10 VHVMs, SF or FF were included in the nutritional quality and marketing analyses. A *p* value of <0.05 was considered statistically significant.

## 3. Results

### 3.1. Prevalence of Foods with Voluntary Nutrient Additions

A total of 52 foods and beverages (<1% total products) were classified as SF (Table 1). Analysis by food subcategory revealed the greatest numbers of SF (Table 1) to be in carbonated and non-carbonated beverages and wine coolers (*n* = 15), and juices, nectars, and fruit drink substitutes (*n* = 24). The most commonly added supplemented ingredients were vitamin B6 (*n* = 20), vitamin B12 (*n* = 15), and niacin/vitamin B3 (*n* = 13). Overall, 13 SFs contained at least one vitamin/mineral that exceeded the RDA or AI (Table 2). Many beverage products also contained high levels of caffeine (Table 3).

Three-hundred and twenty-six (2% of total foods) foods and beverages were classified as FF (Table 1). Food categories containing the greatest number FF were dairy products and substitutes (*n* = 102), bakery products (*n* = 98), and cereals and other grain products (*n* = 76). The most commonly added types of functional ingredients were novel fibres (*n* = 191) and herbals/bio-actives (*n* = 109), inulin (*n* = 134) and probiotic bacterial cultures (*n* = 85) (Table 4).

A total of 923 foods and beverages (6% of total products) were classified as VHVMs. Major food categories containing the greatest number of VHVMs were cereals and other grain products, and fruit juices (Table 1). Food subcategories containing the greatest proportions of VHVMs were ready-to-eat breakfast cereals (puffed and coated, without fruit and nut) (*n* = 59, 77%), fruit juices, nectars, and fruit juice substitutes (*n* = 342, 54%) and vegetable juice and drinks (*n* = 20, 47%). The top five most commonly added vitamins and minerals include vitamin C (*n* = 462), thiamine (*n* = 275), folic acid (*n* = 154), and iron (*n* = 156) (Table 5).

### 3.2. Nutritional Quality of Foods with Voluntary Nutrient Additions

Of the two food categories analyzed (i.e., only subcategories with >10 SF products) only SFs in the juices, nectars and fruit drinks substitutes subcategory had both significantly lower NPSC scores (‘healthier’) and more products meeting ‘healthy’ cut-points than their non-SF counterparts (Table 6).

The median NPSC scores of FFs in four out of ten subcategories examined were significantly ‘healthier’ (i.e., lower scores) compared to non-FFs in these categories with no significant differences amongst FFs and non-FFs in the other subcategories (Table 7). Only two subcategories-grain-based bars (without filling or coating) and juices, nectars and fruit drink substitutes–had a significantly greater proportion of FFs meeting ‘healthy’ cut-points when compared to their non-FF counterparts (Table 7). There was no significant difference in the proportion of products meeting healthy cut-points in the other subcategories examined (Table 7).

Amongst food and beverage subcategories with VHVMs, in four out of eight subcategories analyzed, VHVMs had significantly higher NPSC scores (i.e., were ‘less healthy’) than non-VHVMs (Table 8). In two of these subcategories—hot breakfast cereals and ready-to-eat breakfast cereals, a significantly greater proportions of VHVMs did not meet ‘healthy’ cut-points compared to non-VHVMs (Table 8).

In two out of eight subcategories—plant-based beverages, milk, buttermilk, and milk-based drinks, and pastas without sauce—VHVMs had significantly lower (*p* < 0.05) FSANZ NPSC scores (i.e., were ‘more healthy’) than comparable non-VHVMs. Additionally, a significantly greater proportion of VHVMs in the plant-based beverages, milk, buttermilk, and milk-based drinks subcategory met cut-points to be rated as ‘healthy’, in comparison to non-VHVMs in this subcategory.

There were no statistical differences in median NPSC scores between VHVMs and non-VHVMs in the other food subcategories examined.

### 3.3. Marketing on Food Labels of Nutritionally Enhanced Foods

Irrespective of overall nutritional quality, SFs, FFs and VHVMs were more heavily marketed than comparable foods that did not contain nutritional enhancements (Table 9, Table 10 and Table 11). SFs in both subcategories examined had a greater number of regulated claims (i.e., nutrient content and health claims) than non-SFs. However, unregulated FOP marketing was greater on SFs in comparison to non-SFs only in the juices, nectars, and fruit drink substitutes subcategory (Table 9).

FFs had significantly higher levels of both regulated claims and FOPS in comparison to non-FFs in 7 out of 10 subcategories examined with no significant differences in the remaining 3 (Table 10).

In all but two subcategories, VHVM products carried a significantly greater number of regulated claims (i.e., nutrient content, function, and disease risk reduction claims) than non-VHVMs, and in six out of eight subcategories, a significantly greater number of FOPS occurred on the labels of VHVMs compared to non-VHVMs (Table 11).

## 4. Discussion

This study provides the first comprehensive analysis of the occurrence of nutritionally enhanced foods in the Canadian market and a summary of the types and amounts of nutrients that are added to these foods. It also importantly elucidates the relationship between nutritionally-enhanced products and their overall nutritional quality and promotion (through on package marketing). While nutritionally enhanced foods made up a very small proportion of the total food marketplace, a high concentration of these items was observed in certain foods categories. In most cases, our findings also indicate that such nutritionally enhanced products are not healthier than their unenhanced counterparts, when examined at the food category level.

The indiscriminate addition of vitamin and minerals to foods has garnered considerable concern from researchers and health professionals in recent years given the potential health effects of high nutrient intakes, particularly for certain individuals, including children, that have lower RDAs and ULs [7,14]. Our study found FFs to be most commonly enriched with inulin—a novel and prebiotic fibre—and probiotic bacterial cultures. Consuming high doses of inulin has been associated with short-term gastrointestinal side effects, such as bloating, flatulence, and diarrhea [21], and although probiotics are generally well-tolerated amongst healthy individuals, there are concerns surrounding an increased risk of sepsis if consumed by high-risk populations, such as immunocompromised individuals [22]. FFs were also observed to contain high levels of protein isolates/concentrates, which may not be suitable for consumption among certain individuals, such as those with compromised kidney function [23].

Our study found B vitamins to be the most commonly added micronutrient to SF. As most B vitamins, with the exceptions of niacin/vitamin B3, Folic acid/Vitamin B9 and vitamin B6, do not have defined ULs, manufacturers have wide safety margins to add them to foods in large amounts. The highest amount observed, when compared to the RDA, was vitamin B12 at 5 times the RDA for adults and nearly 7 times the RDA for children. Two foods, however, were found to contain levels of niacin/vitamin B3 that exceeded the UL of adults and twice that of children. In addition to vitamins and minerals, many supplemented beverages contained high levels of caffeine. Our findings support others who observed beverages such as energy drinks to also contain excessive amounts of caffeine [24]. Although maximum levels of caffeine addition to beverages have been set in documents published by Health Canada [8], there is currently no safe level of caffeine consumption that has been established for children and adolescents, and caffeinated beverages have been deemed unsuitable for consumption in these populations [25]. Children and adolescents are at higher risk of developing caffeine toxicity, including serious side effects such as adverse cardiovascular effects, seizures, and even death [26]. Although some SFs, particularly those containing high levels of caffeine, are required to carry cautionary statements [8], such as ‘not recommended for persons under 18 years of age’, these products are easily accessible to all members of the population and sold alongside other beverages. The effectiveness of warning labels in promoting safe consumption is also only now being examined in reference to the Canadian context.

Our analyses of foods with VHVMs, revealed these to be most commonly fortified with vitamin C, thiamine, folic acid/folate, iron, and vitamin D. From the 2004 national nutrition survey (CCHS 2.0), the prevalence of inadequacy for vitamin C, thiamine, folic acid, and iron was very low in Canadian children and adolescents 13 years of age and under [27]. In individuals over the age of 14, the prevalence of inadequate vitamin/mineral intakes were uniformly high only for vitamin A, vitamin D, calcium, and magnesium [27]. Consequently, with the exception of vitamin D, most vitamin and mineral additions appear incongruent with Canadian public health needs. Concerns that the consumption of foods with voluntary nutrient-enhancements may result in excessive nutrient intakes, are further exacerbated when considered in the context of those who also consume these products in combination with supplements [5,14]. Forty-percent of Canadians take some form of vitamin/mineral supplement and data from Canada has shown that the use of supplements is associated with an increased risk of vitamin/mineral intakes above the UL in both children and adults [27], particularly in the cases of folic acid [28], zinc, and iron [27].

In addition to the issues related to excess nutrient exposure from voluntary fortification, there have also been concerns that such a practice may serve to inadvertently promote consumption of foods of otherwise low nutrient quality [5]. While current regulatory frameworks prohibit staple foods from voluntary fortification practice, through standards of identity, food subcategories observed here to engage in voluntary nutrient enhancements can be broadly characterized as having a high degree of product processing (e.g., ready-to-eat breakfast cereals; fruit juices and drinks; cookies and wafers). Our findings also indicated that such products are not necessarily nutritionally superior to similar food products without nutrient additions. Application of The NPSC created by Food Standards Australia New Zealand (FSANZ), revealed SFs had median scores meeting cut-offs to be considered as ‘healthy’ foods. SFs were also either healthier than or as healthy as non-supplemented foods in the subcategories examined, when median NPSC scores were compared between the two groups. FFs, on the other hand, had median NPSC scores that rated them ‘less healthy’ in 4 out of 10 food subcategories analyzed. In 3 out of these 4 subcategories, FFs were still significantly healthier (i.e., had lower median NPSC scores) than non-FFs, despite being rated as ‘less healthy’ foods. This finding, however, may have been driven by the fact that many FFs were observed in product categories generally considered to be less healthy, such as cookies and wafers, and ready-to-eat breakfast cereals.

A starker picture emerged in the comparison of the nutritional quality of foods with VHVM (>25% DV) compared to those without such vitamin and mineral additions. Foods with VHVM were less healthy than foods without, in the majority of food subcategories examined, based on median nutrient profiling scores. These foods, as well as both SFs and FFs, were also found to be more heavily marketed than foods without nutritional-enhancements, irrespective of their nutritional quality scores. Most were marketed using both government regulated (i.e., nutrient content and health claims) and unregulated claims and symbols predominantly found on the front-of-package. While it is to be expected that manufacturers promote their products on the bases of nutritional enhancements, the fact that these products are not necessarily nutritionally superior to products without such nutrient additions is concerning, particularly in the context of a growing body of research confirming nutritional labelling driving product sales [29]. Research from Canada has also indicated that consumers are more likely to include foods of low nutritional value in their diets if they are fortified [30], and that they are more likely to form positive attitudes towards a food—including attitudes regarding the healthiness of a food—if the food label carries nutrition or health-related claims [31].

To prevent the shifting of dietary patterns and displacement of healthy foods, the overall nutritional quality of a food should be taken into consideration in setting regulations for the marketing of discretionary fortified foods. In the United Kingdom, Australia, and New Zealand, nutrient profiling models that take into consideration various nutritional aspects of a food—such as energy, saturated fat, sugar, and sodium content—are used to determine the eligibility of products to carry nutrition-related and health claims. Currently, Health Canada does not use a nutrient profiling system to regulate the use of nutrient content and health claims (with the exception of the few disease-risk reduction claims) nor are there any specific regulations governing the display of unregulated nutritional promotion such as FOP summary systems and symbols.

There are a number of limitations to this study that should be considered in the interpretation of our results. First, the classification of foods as either SFs or FFs was based on definitions specific to this study since Health Canada had not yet provided finalized classification criteria for either of these types of foods at the time these analyses were completed. In addition, a number of foods that would be considered as SFs may not have been captured in this study, because at the time of collection, many SFs had yet to transition from NHPs to be considered as foods under the food regulatory framework and as a result did not carry an NFt on their labels. Only SFs carrying a standardized NFt were included in this study and as such, it is conceivable that the occurrence of SFs in the Canadian food marketplace could have been underestimated. Similarly, while FLIP 2013 represents 75% of the Canadian food and beverage retail market share [17], there still remains a number of products that were not accounted for in the present analysis, some of which may have fallen under our definitions for nutritionally-enhanced products. Furthermore, our study did not include foods for which nutrients were added through biofortification practices. Canada does not have regulations surrounding the labelling of biofortification of food items, although these foods have been approved under the novel food regulations. As such, we were unable to identify these products. It is conceivable that there are products available for sale in the Canadian marketplace that have nutrient enhancement through this route and therefore not captured in the current analysis. It should also be noted that this study relied on declared nutrient compistion reported on the NFt, rather than laboratory analyses, which could affect the accuracy of our resuls. Earlier Canadian research for intstance, has shown that 17% of products had NFt declarations that differed by more than 20% from chemically analyzed values [32].

An additional point of consideration is the means by which product healthfulness was derived. Under the FSANZ framework for nutrient profiling, points are scored for products’ fruit/vegetable/nut/legume (FVNL) content. In this study the rank of ingredients in the Ingredient List (since manufacturers are not required to declare quantitative amounts of ingredients). Although the procedure was standardized using specific cut-points, FVNL content is only an estimation. Finally, while the FSANZ nutrient profiling model has been validated in Australia and New Zealand [33,34], the model has not yet been validated for use in Canada. However, as Health Canada has yet to release a nutrient profiling system developed and validated for use in Canada, the FSANZ model was determined to be the most appropriate for use in this context, as it includes both positive and negative nutrients in its calculations.

This study also draws on a nationally representitive dataset of packaged foods collected in 2013 and as such, it is likley that the prevelence of nutritionally enhanced products has increased, given trends in product innovation and expanded opportunities for nutrient enhancements [35,36]. Nevertheless, this work provides an important baseline from which to track the landscape of voluntary nutritional enhacments in the Canadian food supply. To the best of our knowledge, it is also the first study to evaluate the overall nutritional quality and marketing propoensity of such products, important foundational data necessary to inform the development of regulations in a policy space which to date has had minimal regulatory oversight. Our findings in this respect are particularly timely given newly released policy proposals by Health Canada which aim to establish detailed conditions for the use of supplemented ingredients in foods, including permitted supplemental ingredients, the categories of food to which they may be added, the maximum amount that may be added to a food as well as additional requirements on the labelling and advertising of supplemented foods [37].

## 5. Conclusions

Our study provides important baseline data on the occurrence of nutritionally-enhanced products in the Canadian market in 2013 and can be used to track changes that occur over time, as nutritionally-enhanced foods are a fast-growing sector of the food industry [35,36]. Although foods and beverages with voluntary nutrient additions made up a small proportion of the Canadian prepackaged food supply in 2013, the prevalence was found to be high in certain subcategories, heightening the risk of over-consumption, particularly in certain sub-populations, such as children. Furthermore, these foods were more heavily marketed than similar foods and beverages without voluntary nutrient additions, which may further promote their consumption. It is evident from this study that foods with voluntary nutrient additions are not necessarily of greater nutritional quality than other foods. These results can inform further investigations into the frequency of consumption of these products to garner a better understanding of consumer exposure and risk. Our findings also demonstrate a potential need for changes to current labelling regulations to aid consumers in the selection and appropriate use of nutritionally-enhanced foods to allow for their safe consumption.

## Figures and Tables

**Table 1 nutrients-13-03115-t001:** Prevalence of supplemented foods (SFs) *, functional foods (FFs) ** and foods with very high levels of voluntary fortification (VHVMs) *** in the Canadian marketplace in 2013.

Food Category and Subcategory	Total *n*	Supplemented Foods *n* (% of Total Foods)	Functional Foods *n* (% of Total Foods)	Foods with VHVMs *n* (% of Total Foods)
**Bakery products**	1706	-	98 (6)	3 (<1)
Grain-based bars, with filling or coating	106	-	33 (31)	-
Grain-based bars, without filling or coating	100	-	17 (17)	-
Cookies and graham wafers	391	-	22 (6)	-
Bagels, tea biscuits, scones, rolls, buns, croissants, tortillas, soft bread sticks, soft pretzels and corn bread	288	-	6 (2)	-
Crackers, hard bread sticks and melba toast	279	-	5 (2)	-
Bread, excluding sweet quick-type rolls	231	-	5 (2)	-
French toast, pancakes and waffles	59	-	4 (7)	-
Coffee cakes, doughnuts, danishes, sweet rolls, sweet quick-type breads and muffins	123	-	4 (3)	-
Brownies	28	-	2 (7)	-
Pies, tarts, cobblers, turnovers and other pastries	101	-	-	3 (3)
**Beverages**	422	19 (5)	17 (4)	8 (2)
Carbonated and non-carbonated beverages and wine coolers	268	15 (6)	14 (5)	8 (3)
Sports drinks and water	125	3 (2)	2 (2)	-
Coffee	29	1 (3)	1 (3)	-
**Cereals and other grain products**				
Ready-to-eat breakfast cereals, puffed and coated, without fruit or nuts, very high fibre	793 77	1 (<1) 1 (1)	76 (10) 19 (25)	309 (39) 59 (77)
Ready-to-eat breakfast cereals, fruit and nut type, granola, biscuit type cereals	170	-	30 (18)	56 (33)
Pastas without sauce	439	-	23 (5)	154 (35)
Hot breakfast cereals	107	-	4 (4)	40 (37)
**Combination Dishes**				
Not measurable with a cup	521	-	-	2 (<1)
**Dairy products and substitutes**	2020			
Cheese, including cream cheese and cheese spread	453	4 (<1) 2 (<1)	105 (5) 4 (1)	104 (5) -
Plant-based beverages, milk, buttermilk and milk-based drinks, such as chocolate milk	247	2 (1)	14 (6)	97 (39)
Yogurt	233	-	74 (32)	-
Quark, fresh cheese and fresh dairy desserts	99	-	6 (6)	-
Shakes and shake substitutes	11	-	4 (36)	-
Desserts				
Custard, gelatin and pudding	395	-	-	7 (2)
Dairy desserts, frozen (cakes, bars, sandwiches or cones)	187	-	2 (1)	-
Ice cream, ice milk, frozen yogurt and sherbet	395	-	1 (<1)	-
**Fruit and fruit juices**				
Juices, nectars and fruit drink substitutes	636	24 (4)	14 (2)	342 (54)
**Vegetables**				
Vegetable juice and vegetable drink	43	4 (9)	2 (5)	20 (47)
Vegetable juice and vegetable drink				
**Meat, poultry, their products and substitutes**				
Luncheon meats	101	-	1 (1)	-
**Nuts and seeds**				
**Peanut butter**	50	-	1 (2)	-
**Fats and oils**	342		7	
Butter, margarine, shortening and lard	91	-	5 (6)	-
Dressings for salad	252	-	3 (1)	-
**Marine and freshwater animals**				
Marine and fresh water animals without sauce	209	-	4 (2)	-
**Salads**				
Salads, such as egg, fish, shellfish, bean, fruit, vegetable, meat, ham or poultry salad	47	-	-	1 (2)
**Snacks**				
Chips, pretzels, popcorn, extruded snacks, grain-based snack mixes, fruit-based snacks	562	-	-	13 (2)

* Supplemented foods were defined as foods that contain added vitamins, minerals, amino acids, or caffeine added in amounts other than that which is permissible by the current FDR for fortification or enrichment purposes. ** Functional foods were defined as foods that contained substances (other than vitamins and minerals) added for the purpose of providing a health benefit. *** VHVMs were defined as foods containing greater than 25% of the Daily Value (DV) of an added vitamin or mineral in accordance with voluntary fortification policies stated in the Canadian Food and Drug Regulations (3).

**Table 2 nutrients-13-03115-t002:** Levels of added vitamins and minerals in the most common supplemented food * (SF) categories.

Food Category ^¥^	Micronutrient	Number of SFs Containing Nutrient *n* (% of Total SFs/Category)	RDA ^∫^ or AI ^φ^ /UL		Median Amount/ Serving	Amount/Serving (%DV)	
			Adult	Child		Min	Max
Juices, nectars and fruitdrink substitutes	Folic acid (µg)	7 (29)	400/1000	300/600	99	66 (30)	132 (60)
	Riboflavin (mg)	2 (8)	1.3/ND	0.9/ND	0.1	0.1 (6)	0.1 (6)
	Thiamine (mg)	9 (38)	1.2/ND	0.9/ND	0.2	0.1 (8)	0.2 (15)
	Vitamin B6 (mg)	2 (8)	1.3/100	1.0/60	0.12	0.12 (6)	0.12 (6)
	Vitamin D (IU)	9 (38)	600/4000	600/4000	100	16 (8)	120 (60)
	Vitamin E (mg)	5 (21)	15/1000	11/600	2	2 (20)	2.5 (25)
	Calcium (mg)	10 (42)	1000/2500	1300/3000	330	88 (8)	330 (30)
	Iron (mg)	6 (25)	8/45	8/40	1.4	1.4 (10)	2.1 (15)
	Magnesium (mg)	2 (8)	400/350	240/350	25	25 (10)	25 (10)
	Potassium (mg)	4 (17)	4700 ^φ^/ND	4500 ^φ^/ND	297.5	245 (7)	385 (11)
Carbonated and non-carbonated beverages and wine coolers	Niacin (mg)	11 (61)	16/35	12/20	20.7	5.8 (25)	39.1 (170)
	Pantothenic acid (mg)	11 (61)	5 ^φ^/ND	4 ^φ^/ND	4.9	1.8 (25)	20.3 (290)
	Vitamin B6 (mg)	15 (83)	1.3/100	1/60	4.0	0.5 (25)	7.0 (390)
	Vitamin B12 (µg)	12 (67)	2.4/ND	1.8/ND	8.1	5 (250)	12 (600)
	Vitamin C (mg)	2 (11)	90/2000	45/1200	78	100 (100)	160 (160)
	Calcium (mg)	1 (6)	1000/2500	1300/3000	330	330 (30)	330 (30)
	Zinc (mg)	4 (22)	11/40	8/23	0.9	0.9 (10)	3.6 (40)

* Supplemented foods were defined as foods that contain added vitamins, minerals, amino acids, or caffeine added in amounts other than that which is permissible by the current FDR for fortification or enrichment purposes. ^¥^ Food subcategories containing at least 10 supplemented foods are shown. ^∫^ RDAs, AIs, and ULs are based on the highest requirements in adults (excluding pregnant and lactating women, and individuals over the age of 70) and children aged 4–13. ^φ^ Adequate Intakes (AIs) are followed by ^φ^. Abbreviations–SF = supplemented food; RDA = Recommended Dietary Allowance; AI = Adequate Intake; UL = Upper Limit; % DV = Percent Daily Value (3).

**Table 3 nutrients-13-03115-t003:** Added caffeine and amino acids in supplemented foods * (SFs) in Canada in 2013.

Food Category	Micronutrient	Number of SFs Containing Nutrient *n* (% of Total SFs)	Max. Amount Permitted ^¥^	Amount/Serving		
				Median	Min	Max
Carbonated and non-carbonated beverages and wine coolers	Taurine (mg/serving)	11 (61)	3000	1000	200	2000
	L-theanine (mg/serving)	2 (11)	300	25	25	25
	Caffeine (ppm)	11 (61)	400	320	320	360

* Supplemented foods were defined as foods that contain added vitamins, minerals, amino acids, or caffeine added in amounts other than that which is permissible by the current FDR for fortification or enrichment purposes. ^¥^ Maximum levels of addition for caffeine and amino acids are set out in Guidance Documents pertaining to products that have been issued TMAs. Abbreviations–SF = supplemented food.

**Table 4 nutrients-13-03115-t004:** Ingredients added to functional foods * in the Canadian food market in 2013.

	Number of FFs ^¥^ Containing Ingredient per Food Category *n* (% of Total FFs in Food Category)				
Ingredient	Bakery Products (*n* = 98)	Beverages (*n* = 17)	Cereals and Other Grains (*n* = 76)	Dairy Products and Substitutes (*n* = 102)	Fruit and Fruit Juices (*n* = 14)
**Novel fibres**					
Inulin	78 (80)	2 (12)	36 (47)	11 (11)	4 (29)
Corn bran	7 (7)	-	14 (18)	-	-
Wheat bran	38 (39)	-	25 (33)	1 (1)	-
Oat bran	9 (9)	-	2 (3)	-	-
Oat hull fibre	-	-	6 (8)	-	-
Beta-glucan	1 (1)	-	-	-	-
Acacia gum	-	-	1 (1)	-	4 (29)
Polydextrose	-	-	1 (1)	-	-
Dextrin	-	-	1 (1)	-	1 (7)
Psyllium seed husk	-	-	2 (3)	-	-
**TOTAL:**	92 (94)	2 (12)	74 (97)	11 (11)	8 (57)
**Herbals/bioactives**					
Probiotic cultures	3 (3)	-	2 (3)	79 (78)	1 (7)
Ginseng extract	-	7 (41)	-	-	-
Yerba mate extract	-	1 (6)	-	-	-
Milk thistle seed extract	-	2 (12)	-	-	-
Green tea extract	3 (3)	2 (24)	1 (1)	-	-
Guarana seed extract	-	6 (35)	-	-	-
Green coffee bean extract	-	3 (18)	-	-	-
Bee pollen	-	-	1 (1)	-	-
Maca root	-	-	1 (1)	-	-
Red wine extract	1 (1)	-	-	-	-
Rosemary extract		-	1 (1)	-	-
**TOTAL:**	7 (7)	16 (94)	6 (8)	79 (78)	1 (7)
**Omega-3/omega-6**					
DHA oil	-	-	-	1 (1)	-
Encapsulated fish oil	-	-	-	4 (4)	2 (14)
Flaxseed oil	-	-	-	11 (11)	1 (7)
**TOTAL:**	-	-	-	15 (15)	3 (21)
**Protein concentrates/isolates**					
Soy protein	7 (7)	-	6 (8)	1 (1)	-
Whey protein	4 (4)	1 (6)	-	4 (4)	-
**TOTAL:**	11 (11)	1 (6)	6 (8)	4 (4)	-
**Other novel ingredients**	-	-	-	-	3 (21)
Plant sterols	-	-	-	-	3 (21)

* Functional foods were defined as foods that contained substances (other than vitamins and minerals) added for the purpose of providing a health benefit. ^¥^ Major food categories containing at least 10 functional foods (FFs) are shown. Abbreviations–FF = functional food.

**Table 5 nutrients-13-03115-t005:** Vitamins and minerals present in VHVMs * in the Canadian market in 2013.

Food Category ^¥^	Vitamin/Mineral	Number of VHVMs Containing Nutrient (% of Total VHVMs)
Cereals and other grain products	Folic acid/folate	154 (50)
	Thiamine	275 (89)
	Iron	156 (51)
	Magnesium	3 (<1)
	Niacin/niacinamide	2 (<1)
Dairy products and substitutes juices Vegetables ^†^	Vitamin B12	90 (92)
	Calcium	84 (86)
	Vitamin D	98 (100)
	Riboflavin	2 (2)
	Vitamin C	462 (100)
	Vitamin C	20 (100)

* VHVMs were defined as foods containing greater than 25% of the Daily Value (DV) (3) of an added vitamin or mineral in accordance with voluntary fortification policies stated in the Canadian Food and Drug Regulations (FDR). ^¥^ Major food categories containing at least 10 VHVMs were analyzed. ^†^ Fortified foods in the ‘Vegetable’ category include processed, prepackaged vegetable products (e.g., canned green beans).

**Table 6 nutrients-13-03115-t006:** Comparison of NPSC nutrient profile scores * of supplemented foods ^¥^ (SFs) and non-supplemented foods (non-SFs) and the number and proportion of these meeting ‘healthy’ cutpoints in the Canadian marketplace.

Food Category	SFs (*n*)	Non-SFs (*n*)	Median Nutrient Profiling Score (Q1, Q3) ^‡^		*p*-Value ^ψ^	Direction of Significance	Number of Foods Meeting ‘Healthy’ Cutpoints (%)		*p*-Value ^ψ^
			SF	Non-SF			SF	Non-SF	
Carbonated and non-carbonated beverages and wine coolers	15	253	0 (−2, 2)	1 (0, 2)	0.52	NS	8 (53)	104 (41)	0.35
Juices, nectars and fruit drink substitutes	24	612	−2 (−3, 0)	0 (−2, 1)	0.02	Positive	20 (83)	379 (62)	0.03

* Nutrient profile scores are based on the FSANZ Nutrient Profiling Scoring Criterion (NPSC). ^¥^ Supplemented foods (SFs) were defined as foods containing added vitamins, minerals, amino acids, or caffeine added in amounts beyond what is permissible for fortification/enrichment purposes according to the FDR. Food subcategories containing at least 10 SFs were analyzed. ^‡^ Q1 and Q3 refer to 25% and 75% quartile median values. ^ψ^ A *p*-value < 0.05 was considered significant. Positive significance indicates that median scores of SFs are significantly lower (or “healthier”) than median scores of non-SFs. NS indicates no significance. Products meeting specific cut points (beverages: <1; cheese, edible oil, edible oil spreads, margarine, or butter: <28; all other foods: <4) based on the NPSC system are considered ‘healthy’ and would be permitted to carry a health claim in Australia/New Zealand.

**Table 7 nutrients-13-03115-t007:** Comparison of NPSC nutrient profile scores * of functional foods ^¥^ (FFs) and non-functional foods (non-FFs) and the number and proportion of these meeting ‘healthy’ cutpoints in the Canadian marketplace.

Food Subcategory	FFs (*n*)	Non-FFs (*n*)	Median Nutrient Profiling Score (Q1, Q3) ^‡^		*p*-Value ^ψ^	Direction of Significance	Number of Foods Meeting ‘Healthy’ Cutpoints (%)		*p*-Value ^ψ^
			FF	Non-FF			FF	Non-FF	
Cookies and graham wafers	22	369	11 (9, 13)	20 (15, 23)	<0.0001	Positive	0 (0)	3 (1)	0.67
Grain-based bars (with filling or coating)	33	73	11 (9, 12)	16 (10, 19)	0.0002	Positive	0 (0)	2 (3)	0.34
Grain-based bars (without filling or coating)	17	83	4 (2, 9)	9 (8, 11)	0.002	Positive	8 (47)	5 (6)	<0.0001
Carbonated and non-carbonated beverages and wine coolers	14	254	0 (0, 2)	1 (0, 2)	0.24	NS	8 (57)	104 (41)	0.23
RTE breakfast cereals (puffed, coated, w/o fruit or nut, very high fibre)	19	58	10 (8, 13)	12 (3, 14)	0.28	NS	4 (21)	15 (26)	0.67
RTE breakfast cereals (fruit and nut type, granola, biscuit type)	30	140	2 (1, 10)	1 (−1, 8)	0.09	NS	18 (60)	92 (66)	0.55
Pastas without sauce	23	416	−6 (−6, −6)	−4 (−4, −3)	<0.0001	Positive	23 (100)	413 (99)	0.68
Plant-based beverages, milk, buttermilk and milk-based drinks	14	233	0 (−1, 2)	−1 (−1, 0)	0.1	NS	9 (64)	175 (75)	0.37
Yogurt	74	159	0 (−2, 2)	−1 (−2, 1)	0.09	NS	70 (95)	150 (94)	0.94
Juices, nectars, and fruit drink substitutes	14	622	0 (0, 0)	0 (−3, 1)	0.67	NS	13 (93)	388 (62)	0.02

* Nutrient profile scores are based on the FSANZ Nutrient Profiling Soring Criterion (NPSC). ^¥^ Functional foods (FFs) were defined as foods containing added ingredients other than vitamins and minerals for the purpose of providing a health benefit. Food subcategories containing at least 10 FFs were analyzed. ^‡^ Q1 and Q3 refer to 25% and 75% quartile median values. ^ψ^ A *p*-value < 0.05 was considered significant. Positive significance indicates that median scores of FFs are significantly lower (or “healthier”) than median scores of non-FFs. NS indicates no significance in nutrient profiling scores. Products meeting specific cut points (beverages: <1; cheese, edible oil, edible oil spreads, margarine, or butter: <28; all other foods: <4) based on the NPSC system are considered ‘healthy’ and would be permitted to carry a health claim in Australia/New Zealand.

**Table 8 nutrients-13-03115-t008:** Comparison of FSANZ nutrient profiling scores * of VHVMs ^¥^ and non-VHVMs and the number and proportion of these meeting ‘healthy’ cutpoints in the Canadian marketplace.

Food Subcategory	VHVMs (*n*)	Non-VHVMs (*n*)	Median Nutrient Profiling Score (Q1, Q3) ^‡^		*p*-Value ^ψ^	Direction of Significance	Number of Foods Meeting ‘Healthy’ Cutpoints (%) ^Φ^		*p*-Value ^ψ^
			VHVMs	Non-VHVMs			VHVMs	Non-VHVMs	
Hot breakfast cereals	40	67	9 (5, 11)	−5 (−6, −3)	<0.0001	Negative	10 (25)	61 (91)	<0.0001
RTE cereals (puffed and coated, without fruit or nuts, very high fibre)	59	18	13 (10, 15)	0.5 (0, 5)	<0.0001	Negative	6 (10)	13 (72)	<0.0001
RTE cereals (fruit and nut type, granola, biscuit type)	56	114	2 (0, 8)	1 (−1, 8)	0.27	NS	35 (63)	75 (66)	0.67
Pastas without sauce	154	285	−4 (−6, −4)	−4 (−4, −3)	<0.0001	Positive	154 (100)	281 (99)	0.14
Plant-based beverages, milk, buttermilk, and milk-based drinks	97	150	−1 (−2, 0)	0 (−1, 1)	<0.0001	Positive	92 (95)	91 (61)	<0.0001
Juices, nectars, and fruit-juice substitutes	342	294	0 (−1, 1)	0 (−6, 1)	<0.0001	Negative	210 (61)	189 (64)	0.45
Fruits (fresh, canned, or frozen)	120	158	−1.5 (−4, 0)	−2 (−8, 0)	0.002	Negative	120 (100)	156 (99)	0.22
Vegetable juice and drink	20	23	0 (−1, 0.5)	0 (−3, 1)	0.64	NS	15 (75)	16 (70)	0.69

* Nutrient profiling scores were based on the FSANZ NPSC system. ^¥^ VHVMs were defined as foods containing greater than 25% of the Daily Value (DV) (3) of an added vitamin or mineral in accordance with voluntary fortification policies stated in the Canadian Food and Drug Regulations. Food subcategories containing a minimum of 10 VHVMs were analyzed. ^‡^ Q1 and Q3 represent 25% and 75% quartile median values. ^ψ^
*p* < 0.05 was considered significant. Positive significance signifies that median scores VHVMs are significantly lower (or “healthier”) than median score values of non-VHVMs. Negative significance indicates that median scores of VHVMs are significantly higher (or “less healthy”) than non-VHVMs. NS denotes no significance in median score values. ^Φ^ Products meeting specific cut points (beverages: <1; all other foods except for cheese, edible oil, edible oil spreads, margarine, or butter: <4) based on the NPSC system are considered ‘healthy’ and would be permitted to carry a health claim in Australia/New Zealand.

**Table 9 nutrients-13-03115-t009:** Level of nutrition marketing on food labels of supplemented foods * (SFs) and non-supplemented foods (non-SFs) in Canada in 2013.

Food Subcategory	SFs	Non-SFs	Total Marketing Items ^¥^ Mdn (Q1, Q3) ^ψ^		Significance (*p*) ^‡^	Regulated Claims Mdn (Q1, Q3)		Significance (*p*) ^‡^	FOPS Mdn (Q1, Q3)		Significance (*p*) ^‡^
			SFs	Non-SFs		SFs	Non-SFs		SFs	Non-SFs	
Carbonated and non-carbonated beverages and wine coolers	15	253	2 (1, 3)	0 (0, 2)	Positive (0.0005)	1 (1, 3)	0 (0, 1)	Positive (0.0002)	0 (0, 1)	0 (0, 0)	NS(0.094)
Juices, nectars and fruit drink substitutes	24	612	4 (3.5, 5)	2 (1, 3)	Positive (<0.0001)	3 (2, 4)	1 (1, 2)	Positive (<0.0001)	1 (1, 1)	0 (0, 1)	Positive (0.0015)

* Supplemented foods (SFs) were defined as foods containing added vitamins, minerals, amino acids, or caffeine added in amounts beyond what is permissible for fortification/enrichment purposes according to the FDR. Food subcategories containing at least 10 SFs were analyzed. ^¥^ Total marketing items include the total number of regulated claims (i.e., nutrient content claims and health claims) and Front-of-Pack systems (FOPS) (e.g., ‘made with whole grains’) ^ψ^ Mdn represents median values. Q1 and Q3 represent 25% and 75% quartile median values. ^‡^
*p* < 0.05 is considered significant. Positive significance denotes that SFs had a significantly greater number of total marketing/regulated claims/FOPS than non-SFs. NS represents no significance.

**Table 10 nutrients-13-03115-t010:** Level of nutrition marketing on food labels of functional foods * (FFs) and non-functional foods (non-FFs) in Canada in 2013.

Food Subcategory *	FFs (*n*)	Non-FFs (*n*)	Total Marketing Items ^¥^ Mdn (Q1, Q3)		Significance (*p*) ^‡^	Regulated Claims Mdn (Q1, Q3)		Significance (*p*) ^‡^	FOPS Mdn (Q1, Q3)		Significance (*p*) ^‡^
			FFs	Non-FFs		FFs	Non-FFs		FFs	Non-FFs	
Cookies and graham wafers	22	369	2.5 (2, 4)	0 (0,1)	Positive (<0.0001)	2 (1, 3)	0 (0, 1)	Positive (<0.0001)	1 (1, 1)	0 (0, 0)	Positive (<0.0001)
Grain-based bars (with filling or partial or full coating)	33	73	2 (1, 2)	0 (0, 1)	Positive (<0.0001)	1 (1,2)	0 (0,1)	Positive (<0.0001)	1 (0, 1)	0 (0, 0)	Positive (<0.0001)
Grain-based bars (without filling or coating)	17	83	2 (2, 3)	1 (1, 2)	Positive (0.0008)	2 (1, 2)	1 (0, 2)	NS (0.06)	1 (0, 1)	0 (0, 1)	Positive (0.02)
Carbonated and non-carbonated beverages and wine coolers	14	254	2 (0, 3)	0 (0, 2)	Positive (0.02)	1 (0, 3)	0 (0, 1)	Positive (0.01)	0 (0, 1)	0 (0, 0)	NS (0.06)
Ready-to-eat breakfast cereals (puffed and coated, without fruit or nut, very high fibre)	19	58	6 (5, 7)	3 (1, 4)	Positive (<0.0001)	4 (3, 5)	2 (1, 3)	Positive (0.0004)	2 (1, 2)	1 (0, 1)	Positive (0.0005)
Ready-to-eat breakfast cereals (fruit and nut type, granola, biscuit-type)	30	140	5 (4, 7)	4 (2, 6)	Positive (0.03)	3.5 (2, 6)	3 (1, 5.5)	NS (0.11)	1 (1, 2)	1 (0, 1)	Positive (0.01)
Pastas without sauce	23	416	3 (3, 4)	0 (0, 2)	Positive (<0.0001)	2 (1, 3)	0 (0, 2)	Positive (<0.0001)	2 (1, 2)	0 (0, 0)	Positive (<0.0001)
Plant-based beverages, milk, buttermilk, and milk-based drinks	14	233	3 (3, 5)	2 (1, 4)	Positive (0.02)	3 (3, 5)	2 (1, 4)	NS (0.06)	0 (0, 1)	0 (0, 0)	NS (0.26)
Yogurt	74	159	2 (2, 3)	2 (1, 3)	Positive (0.005)	2 (2, 3)	2 (1, 2)	Positive (0.0003)	0 (0, 0)	0 (0, 0)	NS (0.24)
Juices, nectars, and fruit drink substitutes	14	622	4 (3, 5)	1 (0, 2)	Positive (<0.0001)	3 (2, 3)	1 (0, 2)	Positive (<0.0001)	1 (1, 2)	0 (0, 1)	Positive (<0.0001)

* Functional foods (FFs) were defined as foods containing substances (other than vitamins and minerals) added for the purpose of providing a health benefit. Food subcategories containing at least 10 FFs were analyzed. ^¥^ Total marketing items include the total number of regulated claims (i.e., nutrient content claims and health claims) and Front-of-Pack systems (FOPS) (e.g., ‘with whole grains’). Mdn represents median values. Q1 and Q3 represent 25% and 75% quartile median values. ^‡^
*p* < 0.05 is considered significant. Positive significance denotes that FFs had a significantly greater number of total marketing/regulated claims/FOPS than non-FFs. NS represents no significance.

**Table 11 nutrients-13-03115-t011:** Levels of nutrition marketing on VHVMs * compared to non-VHVMs in Canada in 2013.

Food Subcategory	VHVMs (*n*)	Non-VHVMs (*n*)	Total Marketing Items Mdn (Q1, Q3) ^¥^		Significance (*p*) ^‡^	Regulated Claims Mdn (Q1, Q3) ^¥^		Significance (*p*) ^‡^	FOPS Mdn (Q1, Q3) ^¥^		Significance (*p*) ^‡^
			VHVMs	Non-VHVMs		VHVMs	Non-VHVMs		VHVMs	Non-VHVMs	
Hot breakfast cereals	40	67	5 (3, 5)	1 (0, 3)	Positive (<0.0001)	3 (2, 4)	1 (0, 3)	Positive (<0.0002)	1 (1, 2)	0 (0, 1)	Positive (<0.0001)
Ready-to-eat breakfast cereals (puffed and coated, without fruit or nuts, very high fibre)	59	18	4 (2, 6)	2 (1, 4)	Positive (0.02)	3 (2, 4)	2 (1, 3)	Positive (0.007)	1 (0, 2)	1 (0, 1)	NS (0.30)
Ready-to-eat breakfast cereals (fruit and nut type, granola, biscuit type cereals)	56	114	4 (2, 6)	3 (2, 4)	Positive (0.04)	3 (1.5, 4)	2 (1, 4)	NS (0.20)	1 (0, 2)	0.5 (0, 1)	Positive (0.002)
Pastas without sauce	154	285	1 (0, 3)	0 (0, 1)	Positive (<0.0001)	1 (0, 2)	0 (0, 1)	Positive (<0.0001)	0 (0, 1)	0 (0, 0)	Positive (<0.0001)
Plant-based beverages, milk, buttermilk, and milk-based drinks	97	150	4 (3, 5)	2 (1, 3)	Positive (<0.0001)	4 (3, 5)	2 (1, 3)	Positive (<0.0001)	0 (0, 1)	0 (0, 0)	Positive (0.0009)
Juices, nectars, and fruit-juice substitutes	342	294	2 (1, 3)	1 (0, 2)	Positive (<0.0001)	2 (1, 2)	1 (0, 2)	Positive (<0.0001)	1 (0, 1)	0 (0, 1)	Positive (0.0006)
Fruit (fresh, canned, or frozen)	120	158	2 (0, 3)	1 (0, 2)	Positive (0.03)	0 (0, 1)	0 (0, 2)	NS	1 (0, 1)	0 (0, 1)	Positive (0.0003)
Vegetable juice and vegetable drink	20	23	4.5 (2.5, 6.5)	1 (0, 4)	Positive (0.0005)	4 (2.5, 4)	1 (0, 3)	Positive (0.0005)	1 (1, 2)	1 (0, 1)	NS (0.08)

* VHVMs were defined as foods containing greater than 25% of the Daily Value (DV) of an added vitamin or mineral in accordance with voluntary fortification policies stated in the Canadian Food and Drug Regulations. Food subcategories containing at least 10 VHVMs were analyzed. ^¥^ Mdn represents median values; Q1 and Q3 represent 25% and 75% quartile median values. Total marketing items include the total number of regulated claims (i.e., nutrient content claims and health claims) and Front-of-Pack systems (FOPS) (e.g., ‘with whole grains’). ^‡^ *p* < 0.05 was considered significant. Positive significance indicates that VHVMs have a significantly higher median number of total marketing items/regulated claims/FOP than non-VHVMs. NS denotes no significant differences.

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
