# Peer review of "Examining the Prevalence, Nutritional Quality and Marketing of Foods with Voluntary Nutrient Additions in the Canadian Food Supply"

_nutrients, 2021, doi:10.3390/nu13093115_

Round 1

Reviewer 1 Report

In the paper an overview is presented of foods with voluntarily added nutrients and compounds in Canada. These foods were compared with regular foods to find out if these foods differed in profile (healthy food). In general the paper is well written and easy to follow. Although the question remains if the main concerns about these type of foods (as outlined in the introduction) are sound. The conclusion of the paper is that "these findings support concerns surrounding the consumption of foods with voluntary nutrients", but it is not completely clear on which findings this was based. Levels exceeding the RDA or AI is not per definition a problem, it depends on the intake from other sources as well. In addition, some food-groups with added nutrients and compounds tended to have similar profile as regular foods, others better and others worse. So no clear overall picture can be presented. The impact on the intake or at least a rough indication showing if there would be a potential harmful effect, would improve the paper.

In the paper the 2013 situation is presented. In addition, in the paper it is claimed that the market of foods with added nutrients is growing. It would be interesting to add in the discussion a small insight in the current market. I can imagine, no database is present for 2021, but some insight proving the market growth.

Canadion NFt: first abbreviation is used and there after it is fully written. Please change this.

Can you please explain what is Schedule M of the FDR, for a non Canadian this is not clear.

Can you explain why you excluded foods with added nutrients ≤ 25%DV. Is this based on the Canadian legislation? In Europe the minimum additions are 15% of reference intake for label information. 

Can you please mention in discussion the drawback of using label information on nutrient content. There are several papers showing that the actual chemical content of foods deviates from the label information. Not only overages are present, but also lower amounts are possible. This may effect the conclusions about potential high intakes.

niacin/vitamin B1 do you mean these are synonyms? Which is not true. Niacin =B3 and Thiamin =B1. Or do you mean both have n=13? However also later in paper this way of noting was applied. It would be more clear to remove / and use , or and.

Can you please add what DV (daily value) represents. Also add in footnote of tables.

Table 5 has the category vegetables. How can nutrients be added to vegetables, of do you mean processed vegetable foods? Please be clear.

There are also other ways to increase nutrient content of foods, e.g. biofortification. Is that also used in Canada? Perhaps good to also mention these types of adding nutrients to foods, which are probably not (yet) regulated in Canada, similar to EU.

In the results section FSANZ NPSC scores are presented. But what is also interesting to know is what made these foods better or worse. What aspects of the score were better/worse. This can help to get insight in de quality of the foods.

Several nutrients are mentioned not having an UL, one is folic acid, but that has a UL.

Author Response

In the paper an overview is presented of foods with voluntarily added nutrients and compounds in Canada. These foods were compared with regular foods to find out if these foods differed in profile (healthy food). In general the paper is well written and easy to follow. Although the question remains if the main concerns about these type of foods (as outlined in the introduction) are sound. The conclusion of the paper is that "these findings support concerns surrounding the consumption of foods with voluntary nutrients", but it is not completely clear on which findings this was based. Levels exceeding the RDA or AI is not per definition a problem, it depends on the intake from other sources as well. In addition, some food-groups with added nutrients and compounds tended to have similar profile as regular foods, others better and others worse. So no clear overall picture can be presented. The impact on the intake or at least a rough indication showing if there would be a potential harmful effect, would improve the paper.

RESPONSE: We thank the reviewer for their thoughtful assessment of our study. We have endeavored to provide a more nuanced interpretation of our findings and have amended the line in the abstract referenced by the reviewers above.  The implications of our findings have been captured in detail in the discussion and summarized below.  

This study has revealed that many Supplemented Foods have very high levels of caffeine and that two products specifically had levels of niacin/vitamin B3 that exceeded the UL of adults and twice that of children (Discussion lines 34-41). In most cases, our findings also indicated that nutritionally enhanced products were not healthier than their unenhanced counterparts, when examined at the food category level. This was particularly evident amongst foods with very high levels of voluntary fortification (VHVM), where many categories with VHVM additions were less healthy than foods without, based on median nutrient profiling scores (FSANZ NPSC) (Discussion, lines 92-97). These foods, as well as both Supplemented Foods and Functional Foods, were also found to be more heavily marketed than foods without nutritional-enhancements, irrespective of their nutritional quality scores. Most nutritionally-enhanced foods identified in our study were also more marketed using both government regulated (i.e. nutrient content and health claims) and unregulated claims and symbols predominantly found on the front-of-package. While our study does not specifically examine food sales or consumption, our findings in context with other work which have found food marketing to drive sales and reveal consumer’s perception of nutritionally enhanced foods as being ‘healthier’ than conventional food products (Discussion, lines 98-109), signal potential dietary implications for consumers, and a need to further monitor this practice of voluntary fortification by manufacturers and its impacts product sales and consumption.

In the paper the 2013 situation is presented. In addition, in the paper it is claimed that the market of foods with added nutrients is growing. It would be interesting to add in the discussion a small insight in the current market. I can imagine, no database is present for 2021, but some insight proving the market growth.

RESPONSE: Thank you for your suggestion. We have just finished collecting data for our 2020 dataset in the past year, we look forward to examining the changes to the landscape of voluntary fortified foods. It is likely that the prevalence of nutritionally enhanced products has increased given trends in product innovation and expanded opportunities for nutrient enhancements in Canada, particularly in the context of supplemented foods. A discussion to this end has been included in the Discussion section of the manuscript (lines 157-171).

Canadion NFt: first abbreviation is used and there after it is fully written. Please change this.

RESPONSE: Thank you for alerting us to this error. It has been amended.

Can you please explain what is Schedule M of the FDR, for a non Canadian this is not clear.

RESPONSE: Schedule M is a component of the Canadian Food and Drug Nutrition Labelling Regulations [B.01.001]. This section lists serving size reference amounts and recommended serving size ranges for food categories and subcategories. We have updated our manuscript to include this information (Section 2.1.1).

Can you explain why you excluded foods with added nutrients ≤ 25%DV. Is this based on the Canadian legislation? In Europe the minimum additions are 15% of reference intake for label information. 

RESPONSE: Yes. Foods with Voluntary Fortification were only defined as those with added nutrients >25% DV, in accordance with voluntary fortification policies stated in the Canadian Food and Drug Regulations. This has been clarified in the methods section of our manuscript (Section 2.2.1).

Can you please mention in discussion the drawback of using label information on nutrient content. There are several papers showing that the actual chemical content of foods deviates from the label information. Not only overages are present, but also lower amounts are possible. This may effect the conclusions about potential high intakes.

RESPONSE: Thank you for your comment. We have noted in the limitation section of our manuscript that there are potentially differences between the nutrient content listed on the Nutrition Facts table and the actual compositional quantity of that nutrient in the food. Work from the Canadian context has examined differences between declared nutrient content to that derived from chemical analysis and found nearly 17% of foods had nutrient values that varied by more than 20% from the analyzed value (Fitzpatrick et al, 2014;  ref, 33). These differences were only statistically significant for sodium and calorie content, both of which were underreported on the Nutrition Facts table. As a result, it is conceivable that calculated FSANZ scores may have been higher (‘less healthy’) than those calculated using the Nutrition Facts table and potentially affecting the relationship between the presence of nutrient enhancements and overall nutritional quality.

niacin/vitamin B1 do you mean these are synonyms? Which is not true. Niacin =B3 and Thiamin =B1. Or do you mean both have n=13? However also later in paper this way of noting was applied. It would be more clear to remove / and use , or and.

RESPONSE: Thank you. This was an error and we appreciate the reviewer drawing it to our attention. We have updated our manuscript to correctly refer to the synonyms as Niacin/vitamin B3 throughout.

Can you please add what DV (daily value) represents. Also add in footnote of tables.

RESONSE: A description of what we mean by Daily Value (DV) has been added to the methods section where the term first appears. Footnotes in tables have also been updated to include a reference where this concept is further described.

Table 5 has the category vegetables. How can nutrients be added to vegetables, of do you mean processed vegetable foods? Please be clear.

RESPONSE: We have clarified by including a footnote in Table 5 which describes Fortified foods in the ‘Vegetable’ category as including processed, prepackaged vegetable products (e.g. canned green beans).

There are also other ways to increase nutrient content of foods, e.g. biofortification. Is that also used in Canada? Perhaps good to also mention these types of adding nutrients to foods, which are probably not (yet) regulated in Canada, similar to EU.

RESPONSE: This is a very valid point. While some biofortified foods are approved as novel foods under the novel food regulations the reviewer is correct that Canada does not have regulations surrounding the labelling of biofortification of food items. As such we were unable to identify these products and it is conceivable that there are products available for sale in the Canadian market place that have nutrient enhancement through this route and therefore not captured in the current analysis. This point has been added to the discussion.

In the results section FSANZ NPSC scores are presented. But what is also interesting to know is what made these foods better or worse. What aspects of the score were better/worse. This can help to get insight in de quality of the foods.

RESPONSE: We agree with the reviewer’s suggestion. It would be interesting to know what nutritional qualities drove the individual FSANZ NPSC scores attributed to each food. For the purposes of our current analyses however, the NPSC score provided a picture of a food’s overall nutritional quality in order to investigate the primary aim of our study which was to assess the global ‘healthfulness’ of a food in relation to its voluntary fortification status. The computational demand for elucidating the exact nutritional driver(s) which make the large number of food categories and subcategories ‘healthy’ or not was beyond the scope of this current analysis. Subsequent work, however, which will examine changes to the landscape of voluntary fortified foods will endeavour to provide a more nuanced assessment.

Several nutrients are mentioned not having an UL, one is folic acid, but that has a UL.

RESPONSE: Thank you for drawing this to our attention. We have corrected this in our manuscript (line 33).

Reviewer 2 Report

The paper entitled “Examining the prevalence, nutritional quality and marketing of foods with voluntary nutrient additions in the Canadian food supply” aimed to evaluate the prevalence, nutritional quality, and marketing characteristics of foods with added nutrients in the Canadian market. Although the title is promising, the value of the paper is quite low mostly due to the data used for the analysis. They are likely to be out of date as they come from 2013 and refer to food market that is fast changing, specifically for “novel” foods.

Author Response

The paper entitled “Examining the prevalence, nutritional quality and marketing of foods with voluntary nutrient additions in the Canadian food supply” aimed to evaluate the prevalence, nutritional quality, and marketing characteristics of foods with added nutrients in the Canadian market. Although the title is promising, the value of the paper is quite low mostly due to the data used for the analysis. They are likely to be out of date as they come from 2013 and refer to food market that is fast changing, specifically for “novel” foods.

RESPONSE: We thank the reviewer for their thorough assessment of our study. As noted by the reviewer, our work draws on a nationally representative dataset of packaged foods collected in 2013 and as such it is likely that the prevalence of nutritionally enhanced products has increased given trends in product innovation and expanded opportunities for nutrient enhancements (ref 36 and 37). Nevertheless this work provides an important baseline from which to track the landscape of voluntary nutritional enhancements in the Canadian food supply. More importantly however, a primary aim of this study was to evaluate the overall nutritional quality and marketing propensity of such products, important foundational data necessary to inform the development of regulations in a policy space which to date has had minimal regulatory oversight. Our findings in this respect are particularly timely given newly released policy proposals by Health Canada which aim to establish detailed conditions for the use of supplemented ingredients in foods, including permitted supplemental ingredient, the categories of food to which it may be added, the maximum amount that may be added to a food as well as additional requirements on the labelling and advertising of supplemented foods (38).   This discussion has been added to the manuscript (Discussion; lines 157-171).

Round 2

Reviewer 2 Report

Thank you for your additional explanations. I still have some doubts concerning the currentness and value of the data. I understand that data from 2013 may be important for Canadian readers but are less interesting for general public. Nevertheless, I am giving a positive opinion for the paper to be published in 'Nutrients'.

Some editing corrections are needed. 

A small sugestion - in table 1 add the numbers to main food categories (it will be easily readible information about the whole category; the numbers can be compared between the food categories). 

Author Response

We thank the reviewer for their time and expertise in reviewing our manuscript for a second time. We agree with the suggestion to include the number of products in major food categories in Table 1. We believe this will facilitate ease of comparison at the major category level and as such have including these values for all major categories with more than one subcategory (in order to reduce redundancy in the displayed data).

We have also taken the reviewers suggestion and carefully reedited our manuscript. Corrections have been noted in track-changes.